# Skin Pigmentation Abnormalities and Their Possible Relationship with Skin Aging

**DOI:** 10.3390/ijms22073727

**Published:** 2021-04-02

**Authors:** Ai-Young Lee

**Affiliations:** Department of Dermatology, College of Medicine, Dongguk University Ilsan Hospital, 814 Siksa-dong, Ilsandong-gu, Goyang-si 410-773, Gyeonggi-do, Korea; lay5604@naver.com; Tel.: +82-319617250; Fax: +82-319-617-695

**Keywords:** skin pigmentation abnormalities, skin aging, oxidative stress, DNA damage, telomere shortening, hormonal change, autophagy impairment

## Abstract

Skin disorders showing abnormal pigmentation are often difficult to manage because of their uncertain etiology or pathogenesis. Abnormal pigmentation is a common symptom accompanying aging skin. The association between skin aging and skin pigmentation abnormalities can be attributed to certain inherited disorders characterized by premature aging and abnormal pigmentation in the skin and some therapeutic modalities effective for both. Several molecular mechanisms, including oxidative stress, mitochondrial DNA mutations, DNA damage, telomere shortening, hormonal changes, and autophagy impairment, have been identified as involved in skin aging. Although each of these skin aging-related mechanisms are interconnected, this review examined the role of each mechanism in skin hyperpigmentation or hypopigmentation to propose the possible association between skin aging and pigmentation abnormalities.

## 1. Introduction

Melanin pigmentation plays a critical role in protecting the skin from the harmful effects of ultraviolet (UV) radiation. A variety of genetic and environmental factors are involved in the regulation of pigmentation. Excessive or deficient amounts of melanin cause disorders showing abnormal skin pigmentation. Although wrinkling and laxity are well-identified symptoms [1], abnormal pigmentation is also a common symptom in aging skin [2,3,4,5]. Pigmentation abnormalities associated with skin aging have recently drawn attention, however, not many reports have described the changes in melanin pigmentation in aged human skin, and the role of skin aging in pigmentation abnormalities remains to be clarified.

Human skin undergoes chronological aging and environmental aging. Chronological aging is dependent upon the passage of time. Ultraviolet (UV) irradiation is the primary factor in environmental aging, which is also called photoaging, although other factors such as tobacco smoking and air pollution are involved in environmental aging [6,7]. With age, senescent cells accumulate in human skin, which can compromise skin function and integrity. Chronological aging and photoaging share certain molecular mechanisms (Figure 1). Skin aging is influenced by several factors including oxidative stress, mitochondrial DNA mutations, DNA damage, telomere shortening, and hormonal changes [8,9,10]. Autophagy impairment is also involved in aging and the senescence of skin cells [11,12]. After a brief description of skin pigmentation and skin conditions showing abnormal pigmentation accompanied by aging, this review covers the association between each of the mechanisms implicated in skin aging and skin hyperpigmentation or hypopigmentation to provide a basis for their possible relationship.

## 2. Melanin Pigmentation and Its Abnormalities Accompanied by Skin Aging

Skin pigmentation is largely determined by melanin synthesis in melanocytes, melanosome transfer to keratinocytes, and melanosome degradation. Melanin synthesis is restricted to melanosomes, which contain tyrosinase, the key regulatory enzyme of melanogenesis, and tyrosine-related proteins (TRPs). Tyrosinase is involved in the early steps of melanin synthesis to dopaquinone, which are common to two types of melanin, brown-black eumelanin and yellow-red pheomelanin. Eumelanin is photoprotective by scattering and absorbing UV light. In contrast, pheomelanin is photo-unstable and phototoxic, promoting UV-induced damage including photoaging [13]. Eumelanin and pheomelanin can be produced at different proportions within the same cell. A paucity of eumelanin production makes the skin vulnerable to UV-induced damage [14]. The biogenesis of melanin is controlled by complex regulatory mechanisms. The tyrosinase and TRPs genes have binding sites for microphthalmia-associated transcription factor (MITF), which plays a fundamental role in the transcriptional regulation of melanogenesis via the cAMP, protein kinase C, and nitrogen oxide intracellular signal transduction pathways [15].

Abnormalities in melanin pigmentation are divided into two types, hypermelanosis and hypomelanosis, based on the amount of melanin in the skin. Reduced melanin synthesis with the downregulation of TRP-1, TRP-2, and MITF in senescent melanocytes [16], and decreased melanosome degradation by declining autophagy activity in aging [17] indicate the association of pigmentation abnormalities with skin aging. Their association has also been suggested in inherited and acquired disorders. Inherited disorders showing accelerated aging and skin hyperpigmentation or hypopigmentation provide strong evidence that skin aging can be associated with abnormalities in melanin pigmentation. The favorable therapeutic effects of chemicals and lasers such as polyphenol compounds and fractional photothermolysis, respectively, on both skin aging and pigmentation abnormalities [18,19] also provide reliable evidence for their association. Among the acquired skin disorders of hypermelanosis, melasma is a common skin condition. Although a complex interaction of causative factors is likely to be involved in melasma development [20], a certain proportion of melasma cases are associated with skin aging, either UV-related or UV-unrelated [21,22]. Seborrheic keratosis, which is a benign skin tumor presenting as sharply demarcated light brown-to-black papules containing a greater amount of melanin, develops in elderly individuals, mainly in sun-exposed areas. Senile lentigo is another example of a common benign skin tumor with hyperpigmentation in aged people associated with chronic UV irradiation. As for hypomelanosis, vitiligo is a representative skin condition associated with premature senescence of the skin cells [22]. The role of melanocyte senescence has also been suggested in idiopathic guttate hypomelanosis [23].

## 3. Association of Mechanisms Implicated in Skin Aging with Skin Pigmentation Abnormalities

### 3.1. Role of Oxidative Stress in Skin Pigmentation Abnormalities

Oxidative stress is considered one of the most important mechanisms involved in skin aging [24,25,26,27,28]. Whenever the cellular accumulation of reactive oxygen species (ROS), a family of oxygen-based free radicals including hydrogen peroxide (H_2_O_2_), overcomes the cellular antioxidant capacity, oxidative stress develops. ROS are generated by exogenous stimuli such as UV irradiation [29] and endogenous stimuli including cellular metabolism [30]. The skin is a high turnover, metabolically active organ, requiring energy. Because ROS are produced as the by-products of energy generation by mitochondrial respiration, mitochondria can play a critical role in oxidative stress [28,31]. Conversely, ROS can cause mitochondrial DNA mutations [32], leading to mitochondrial dysfunction. This interconnection provides a reasonable basis for considering mitochondrial DNA mutations in conjunction with oxidative stress. The skin is rich in potential biological targets for oxidative damage such as lipids, proteins, and DNA. Oxidative damage leads to cellular senescence, and thus, skin aging [33,34], therefore, the use of antioxidants can induce anti-aging effects [35,36,37].

#### 3.1.1. Oxidative Stress and Skin Hypermelanosis

The regulatory role of H_2_O_2_ in melanocyte tyrosinase and the amelioration of hyperpigmentation by antioxidants support the effect of oxidative damage in skin hyperpigmentation [38,39,40,41]. The role of oxidative stress in skin pigmentation abnormalities can be predicted by the clinical findings of abnormal skin pigmentation in certain congenital skin diseases caused by mutations in DNA affecting mitochondrial function (Table 1). Mutation of the cytochrome c oxidase subunit 7B (*COX7B*) induces oxidative phosphorylation defects and mitochondrial respiration deficiency [42]. Patients with *COX7B* mutations show eye symptoms and linear skin defects with hyperpigmented streaks [28,43]. Mutations in excision repair cross-complementing group 6 (*ERCC6*) cause defective mitophagy and the accumulation of damaged mitochondria [44]. The *ERCC6* gene encoding a DNA excision repair protein is responsible for Cockayne syndrome group B and can also cause UV-sensitive syndrome, characterized by photosensitivity, hyperpigmentation, freckling, and dryness in sun-exposed areas [45]. A mutation in Fanconi anemia complementation group A (*FANCA*) can reduce the electron transfer between respiratory complex I-III and ROS detoxification enzymes [46,47] and impair mitophagy [48]. Patients with *FANCA* mutation show morphological abnormalities of the skin, including hyperpigmentation [28]. Mutations in the human Harvey rat sarcoma viral oncogene homolog (*HRAS*) can cause mitochondrial dysfunction and defective oxidative phosphorylation [49,50]. *HRAS* mutations and skin hyperpigmentation have been anecdotally associated with the presence of nevi and phacomatosis pigmentokeratotica in patients with the mutation [51,52,53]. Holocytochrome c synthase (HCCS) is an enzyme located in the inner mitochondrial membrane, which adds heme to apocytochrome c and c1. Mutations in *HCCS* mutation reduce oxidative phosphorylation efficiency [54] and cause microphthalmia with linear skin defects syndrome exhibiting skin pigmentation [55]. Protoporphyrinogen oxidase (PPOX), the penultimate enzyme in heme biosynthesis, is a nucleus-encoded flavoprotein associated with the outer surface of the inner mitochondrial membrane. *PPOX* mutations cause mitochondrial abnormalities [56,57,58] and variegate porphyria, which has cutaneous manifestations characterized by photosensitivity, skin fragility, hypertrichosis, and hyperpigmentation [59]. Mutations in *BRAF*, *NRAS*, *KIT*, *GNAQ*, *CDK4*, *PTK2B*, *ERBB4*, *GNA11*, *MEK*, *MITF*, *AKT3*, *MMP*, *CXCR4*, *EPHA3*, *FAS*, *PIK3CA*, *MET*, *CTNNB1*, *NEK10*, and *PDGFRA* have been reported to induce mitochondrial dysfunction, mitochondrial respiratory complex deficiency, and decrease mitochondrial oxidative phosphorylation [60,61], which are associated with skin pigmentation [28]. Mitochondrial DNA polymerase gamma (POLG1) is responsible for mitochondrial DNA replication and repair. *POLG1* mutations result in the depletion of mitochondrial DNA and can cause Alpers-Huttenlocher syndrome, which is characterized by the classic triad of seizures, cognitive degeneration, and hepatopathy. Most patients with this syndrome develop other system involvement [62], such as hyperpigmentation in the antecubital and popliteal fossae and dorsa of the feet [63].

Acquired abnormal skin pigmentation can be related to oxidative stress (Table 2). Sunlight stimulates melanin production for protection against the harmful effects from UV radiation such as from tanning. The substantial role of oxidative stress in UV-induced melanogenesis and the alleviation of UV-induced melanogenesis by antioxidant therapy [64,65,66] indicates an association between oxidative stress and UV-induced hyperpigmentation. UV irradiation can generate the peroxidation products of sebum, squalene monohydroperoxides. The hyperpigmentation of reconstructed human skin equivalents by exposure to squalene monohydroperoxides and the amelioration of squalene monohydroperoxide-induced hyperpigmentation by co-treatment with 12-hydroxystearic acid support a pivotal role of antioxidant mechanisms in UV-induced hyperpigmentation [65]. Although the exact pathogenesis of melasma is uncertain, oxidative stress has been implicated based on increased serum malondialdehyde but reduced serum catalase levels in the patient groups [67,68] and the efficacy of antioxidant drugs identified in a clinical trial [69,70]. The role of oxidative stress has also been identified in seborrheic keratosis. Guanine deaminase, shown to be upregulated in seborrheic keratosis, stimulated UV-induced keratinocyte senescence via ROS generation [71].

#### 3.1.2. Oxidative Stress and Skin Hypomelanosis

Oxidative stress caused by mitochondrial redox imbalance or antioxidant enzyme deficiency is also associated with skin hypopigmentation (Table 1). The loss of ATPase copper-transporting alpha (ATP7A) activity, which causes Menkes disease, results in the accumulation of copper in cellular organelles, including mitochondria, thereby disrupting the mitochondrial redox balance [72]. Tyrosinase is a copper-dependent enzyme involved in melanogenesis. Because ATP7A protein is required for tyrosinase activity, the loss of ATP7A activity by gene mutations impairs melanogenesis [73]. Mutations in SURF1 cytochrome C oxidase assembly factor (*SURF1*) are associated with Leigh syndrome, a rare progressive neurodegenerative disorder caused by mitochondrial cytopathy. Skin and hair abnormalities, including hypopigmentation are often accompanying symptoms [74]. The deletion of mitochondrial DNA induces Kearns-Sayre syndrome, a mitochondrial multisystem disorder characterized by the clinical symptoms of ophthalmoplegia and pigmentary retinopathy. Hypopigmented patches like Hypomelanosis of Ito have been reported as a possible symptom [75]. Piebaldism, characterized by depigmented patches presenting at birth, is caused by *KIT* gene mutations. An association between H_2_O_2_-mediated stress and piebaldism has been identified, although its mechanism in oxidative stress differs from that in vitiligo [76].

Concerning the role of oxidative stress in acquired hypomelanosis, most data are related to vitiligo (Table 2). Oxidative stress caused by ROS production and low levels of antioxidants are considered the pivotal mechanisms in vitiligo development [77,78], although the therapeutic effects of antioxidants have been controversial [79]. Elevations in superoxide dismutase and malondialdehyde [80], reductions in catalase or catalase gene polymorphism [81,82,83], increases in advanced oxidation protein products [84], nuclear factor erythroid 2-like 2 (*Nrf2*) gene polymorphism [85], and reduced Nrf2 activation [86,87] are associated with vitiligo susceptibility. Increased mitochondrial DNA copy numbers [88], mitochondrial dysfunction by sirtuin3 (SIRT3) deficiency [89], and increased calcium influx by the upregulation of transient receptor potential cation channel subfamily M member 2 (TRPM2), a calcium channel sensitive to oxidative stress [90], contribute to oxidative stress-induced melanocyte damage. Autophagy impairment also plays a role in oxidative stress by disrupting the antioxidant defense system [91,92]. ROS accumulation damages melanocytes by the generation of autoantigens, which leads to the CD8+ T cell-mediated destruction of melanocytes.

**Table 1 ijms-22-03727-t001:** Abnormal skin pigmentation in congenital disorders related to oxidative stress.

Pigment Change	Mutated Gene	Dysfunction	Skin Symptom	Reference
Hyper	*COX7B*	Oxidative phosphorylation defect, Mitochondrial respiration deficiency	Linear skin defects with hyperpigmented streaks	[28,42,43]
	*ERCC6*	Accumulation of damaged mitochondria, Mitophagy defect	Photosensitivity, Hyperpigmentation, freckling, and dryness in sun-exposed areas	[44,45]
	*FANCA*	Reduction of electron transfer between respiratory complex I-III and ROS detoxification enzymes, Mitophagy impairment	Morphological abnormalities including hyperpigmentation	[28,46,47,48]
	*HRAS*	Mitochondrial dysfunction, Oxidative phosphorylation defect	Nevi Phacomatosis pigmentokeratotica	[49,50,51,52,53]
	*HCCS*	Reduction of oxidative phosphorylation efficiency	Linear skin defects syndrome exhibiting skin pigmentation	[54,55]
	*PPOX*	Mitochondrial abnormalities	Variegate porphyria characterized by photosensitivity, skin fragility, hypertrichosis, and hyperpigmentation	[56,57,58,59]
	*POLG1*	Mitochondrial DNA depletion	Hyperpigmentation in the antecubital and popliteal fossae and dorsa of the feet	[62,63]
Hypo	*ATP7A*	Mitochondrial redox imbalance by copper accumulation	Melanogenesis impairment	[72,73]
	*SURF1*	Mitochondrial cytopathy	Skin and hair abnormalities including hypopigmentation	[74]
	Mitochondrial DNA deletion	Mitochondrial multisystem disorder	Hypopigmented patches like hypomelanosis of Ito	[75]
	*KIT*	H_2_O_2_-mediated stress	Piebaldism	[76]

**Table 2 ijms-22-03727-t002:** Acquired abnormal skin pigmentation related to oxidative stress.

Pigment Change	Disease/Condition	Evidence for Oxidative Stress-Related	Reference
Hyper	UV-induced melanogenesis	Improvement of hyperpigmentation by antioxidant therapy	[64,65,66]
		Hyperpigmentation on skin equivalents by squalene monohydroperoxides, which is ameliorated by 12-hydroxystearic acid	[65]
	Melasma	Increased malondialdehyde with decreased catalase in serum	[67,68]
		Clinical trial using antioxidant drugs	[69,70]
	Seborrheic keratosis	Role of guanine deaminase in UV-induced keratinocyte senescence via ROS generation	[71]
Hypo	Vitiligo	Elevated superoxide dismutase and malondialdehyde	[80]
		Reduced catalase or catalase gene polymorphism	[81,82,83]
		*Nrf2* gene polymorphism or reduced Nrf2 activation	[85,86,87]
		Increased mitochondrial DNA copy numbers or mitochondrial dysfunction	[88,89]
		Upregulated TRPM2	[90]
		Autophagy impairment	[91,92]

### 3.2. Role of DNA Damage in Skin Pigmentation Abnormalities

DNA repair mechanisms such as the repair of nucleotide excision and double-strand breaks should be activated to maintain homeostasis whenever DNA damage occurs. The downregulation of DNA repair and the accumulation of DNA damage drives the aging process [93,94]. Impaired DNA replication and repair by mutations in a group of premature aging syndromes known as progeroid syndromes provide strong evidence for the role of DNA damage in aging [95]. UV radiation is a notorious inducer of DNA damage in all skin types despite its inverse relationship with constitutive skin pigmentation [96].

#### 3.2.1. DNA Damage and Skin Hypermelanosis

The association between DNA damage and hyperpigmentation, particularly aging-related effects, has been identified in certain DNA repair syndromes caused by pathogenic variants encoding proteins in DNA replication and cellular responses to DNA damage (Table 3). Mutations in RecQ protein-like 4 (*RECQL4*) belonging to the RECQ DNA helicase family, which participate in many aspects of DNA metabolism [97], cause Rothmund-Thomson syndrome [98]. The syndrome shows clinical features of accelerated aging, such as atrophic skin and pigment changes [97]. Mutations in one of the eight genes involved in nucleotide excision repair and DNA polymerase cause Xeroderma pigmentosum [99]. Xeroderma pigmentosum is characterized by the extreme sensitivity to sunlight, resulting in sunburn, unusually increased numbers of lentigines in sun-exposed areas, areas containing both hyperpigmentation and hypopigmentation in the absence of rigorous sunlight protection, accelerated photoaging, and increased skin cancer risk over time [98]. Mutations in several genes involved in DNA repair, particularly interstrand crosslink repair and telomere maintenance, cause Fanconi anemia. A poikilodermatic change with hypopigmentation, hyperpigmentation, and telangiectasia appears in the skin of patients with Fanconi anemia [100,101]. Inactivating mutations of the tumor suppressor serine-threonine kinase 11/liver kinase B1 (*STK11/LKB1*), which can play a role in the regulation of the UV-induced DNA damage response in mice skin [102], underlie Peutz-Jeghers syndrome. Hyperpigmentation of mucous membranes and the skin is one of the characteristic symptoms of Peutz-Jeghers syndrome.

Acquired abnormal skin pigmentation can be related to DNA damage (Table 4). Skin keratinocytes with DNA damage induced by UV exposure secrete α–melanocyte-stimulating hormone (α-MSH) [103]. α-MSH interacts with melanocortin 1 receptor (MC1R), which enhances nucleotide excision repair in melanocytes [104]. The role of α-MSH in melanogenesis suggests an association between DNA damage and melanogenesis. Defects in DNA repair have not been reported in melasma, seborrheic keratosis, and senile lentigo, although microsatellite instability indicating DNA repair defects was detected in seborrheic keratosis developed in a patient with hereditary nonpolyposis colorectal cancer [105].

**Table 3 ijms-22-03727-t003:** Abnormal skin pigmentation in congenital disorders related to DNA damage.

Pigment Change	Mutated Gene	Dysfunction	Skin Symptom	Reference
Hyper	*STK11* /*LKB1*	Abnormal regulation of UV-induced DNA damage response	Hyperpigmentation of mucous membranes and the skin	[102]
Hyper and hypo	*RECQL4*	Defects in many aspects of DNA metabolism	Accelerated aging such as atrophic skin and pigment changes	[97,98]
	*XPA XPB*, *XPC*, *XPD*, *XPE*, *XPF*, *XPG*, or *XPV*	Defects in damaged DNA repair	Photosensitivity, Lentigines, Hyperpigmentation and hypopigmentation, Accelerated photoaging	[98,99]
	Fanconi anemia	Defects in interstrand crosslink repair and telomere maintenance	Poikilodermatic change with hypopigmentation, hyperpigmentation, and telangiectasia	[100,101]
Hypo	Deletion on 15q11.2-q13	Leukocyte telomere length shortening	Hypopigmentation	[106,107]
	Oculocutaneous albinism and Hermansky-Pudlak syndrome	Increased tyrosinase degradation through ubiquitin-proteasome system	Oculocutaneous Albinism, Photoaging	[108,109,110,111]

**Table 4 ijms-22-03727-t004:** Acquired abnormal skin pigmentation related to DNA damage.

Pigment Change	Disease/Condition	Evidence for DNA Damage-Related	Reference
Hyper	UV-induced melanogenesis	Role of α-MSH secreted from UV-damaged keratinocytes in melanogenesis	[103,104]
Hypo	Leukotrichia	Associated with *APE1* polymorphism	[112]

#### 3.2.2. DNA Damage and Skin Hypomelanosis

Certain inherited disorders caused by DNA damage can exhibit aging-related skin hypopigmentation alone or both hypopigmentation and hyperpigmentation (Table 3). As described above, xeroderma pigmentosum patients have hypopigmented areas in addition to hyperpigmented skin [98]. Fanconi anemia shows poikilodermatic changes consisting of hypopigmentation, hyperpigmentation, atrophy, and telangiectasia as the characteristic skin symptoms [100,101]. Poikilodermatic changes also appear in the skin of Rothmund-Thomson syndrome patients [98]. A chromosomal deletion on 15q11.2-q13, which induces shortening of leukocyte telomere length, a marker of biological age, causes Prader-Willi syndrome [106]; as one of the premature aging syndromes, Prader-Willi syndrome patients can exhibit hypopigmentation [107]. Oculocutaneous albinism and Hermansky-Pudlak syndrome are hereditary diseases with DNA mutations [108,109]. Both diseases exhibit oculocutaneous albinism caused by the increased degradation of tyrosinase through the ubiquitin-proteasome system [110]. Photoaging is accelerated in albino hairless mice by exposure to solar simulation [111].

DNA damage and defects in DNA repair have not been identified in vitiligo, although an association between apurinic/apyrimidinic endonuclease 1 (*APE1*) polymorphism with leukotrichia suggests the role of this polymorphism in vitiligo [112] (Table 4).

### 3.3. Role of Telomere Shortening in Skin Pigmentation Abnormalities

Telomeres are dynamic nucleoprotein-DNA structures composed of long tandem TTAGGG repeat, guanine-rich sequences located at the end of chromosomes to cap and protect the linear chromosome ends. Although the length of telomeres differs between chromosome arms, telomeres shorten by cell division, and the initial length of telomeres correlates with the cellular replicative capacity. Accelerated aging in short telomere syndromes caused by inheritable gene mutations resulting in decreased telomere lengths [113] indicates that telomere shortening can be one of the primary hallmarks of aging [114].

#### 3.3.1. Telomere Shortening and Skin Hypermelanosis

The genetically inherited diseases with much shorter telomeres compared to age-matched controls are referred to as telomeropathies. Primary telomeropathies are defined as disorders caused by mutations in the telomere maintenance machinery. Secondary telomeropathies are caused by mutations in genes encoding proteins involved in telomere DNA repair, leading to telomere aberrations and loss [115]. However, overlap exists between telomeropathies, particularly secondary telomeropathies, and DNA mutations.

The role of telomeres in skin pigmentation has been identified in telomeropathies (Table 5). The first disorder identified as a primary telomeropathy was dyskeratosis congenita, which is caused by mutations in the *DKC1* gene encoding the dyskerin protein, or one of the core telomerase genes encoding a protein involved in telomere maintenance [116], resulting in telomere shortening, and thereby, premature aging [117,118]. Reticular hyperpigmentation is a symptom of the dyskeratosis congenita diagnostic triad along with oral leukoplakia and nail dystrophy. Xeroderma pigmentosum and Fanconi anemia are included in secondary telomeropathies, indicating an overlap between telomere shortening and DNA repair syndromes. Among the eight complementation groups in xeroderma pigmentosum, mutations in complementation group C (*XPC*), can cause the instability of telomere upon UV exposure [117]. Fanconi anemia is caused by genetic mutations involved in DNA repair for telomere maintenance. As described before, patients with these disorders manifest abnormal skin pigmentation [100,101]. A point mutation in the *LMNA* gene, which produces abnormal lamin A protein associated with rapid telomere erosion, can cause Hutchinson-Gilford progeria syndrome [119,120]. As one of the most severe segmental progeroid syndromes, patients with Hutchinson-Gilford syndrome demonstrate premature aging with distinct dermatologic features including hyper- and hypopigmentation over areas of sclerodermoid changes [95,121].

Although telomere attrition contributes to certain age-related disorders, the association of melasma, seborrheic keratosis, or senile lentigo with telomere shortening has not been reported.

#### 3.3.2. Telomere Shortening and Skin Hypomelanosis

Telomere shortening can be associated with skin hypomelanosis, although hypopigmentation usually coexists with hyperpigmentation as shown in patients with xeroderma pigmentosum and Hutchinson-Gilford progeria syndrome [95,98] (Table 5).

In contrast, in vitiligo, telomerase activity is lower in lesional skin compared to non-lesional skin and healthy control skin, which was proposed as a reason for the low incidence of sun damage and cancer in the lesional skin of vitiligo patients [122].

### 3.4. Role of Hormones in Skin Pigmentation Abnormalities

Cutaneous neuroendocrine systems can regulate skin function in response to environmental stresses, such as UV radiation and pollutants. Melatonin belongs to the family of neurohormones [123]. The pineal gland is the main organ of melatonin synthesis and secretion. However, human skin can produce and rapidly metabolize melatonin to maintain homeostasis against environmental stresses [124]. Melatonin is involved in the regulation of various physiological activities, including circadian rhythm, immune responses, oxidative process, apoptosis, and mitochondrial homeostasis. Antioxidant actions of melatonin protect skin from UVB-induced harm [125,126], indicating the critical role of melatonin in photoaging.

Receptors for sex steroids, particularly estrogen receptors, are present in human skin, including epidermal keratinocytes and dermal fibroblasts. Many findings, such as the gradual decline of these receptors with chronological and environmental aging [127,128], skin aging accelerated by the loss of estradiol production after menopause, and UV-induced collagen degradation reversed by estradiol replacement [128,129,130], indicate the role of estrogens in protecting skin from aging.

#### 3.4.1. Hormones and Skin Hypermelanosis

Oxidative stress is the main mechanism involved in skin aging and pigmentation. The antioxidant actions of melatonin [125,126] are implicated in the association between melatonin and skin pigmentation. Although significantly lower serum levels of melatonin have been seen in melasma patients compared to controls [68], relevant reports are rare.

Sex hormones, particularly estrogens, play a role, which is stimulatory but not inhibitory, in pigmentation in cultured skin cells [131] and in melasma [20,132]. These results suggest that estrogens enhance skin pigmentation and delay rather than accelerate skin aging.

#### 3.4.2. Hormones and Skin Hypomelanosis

The association of melatonin with vitiligo has been described [133]. A connection between reduced testosterone due to secondary hypogonadism and vitiligo has also been suggested [134].

Above all, vitiligo patients frequently suffer from serious psychological problems such as depression and anxiety. The levels of dehydroepiandrosterone sulfate, which has antioxidant properties, were lower but the ratios of cortisol to dehydroepiandrosterone sulfate were higher in vitiligo patients compared to healthy controls [135]. The results suggest the role of dehydroepiandrosterone sulfate in vitiligo development and provide reliable evidence for the association of vitiligo with stress-related hormones.

### 3.5. Role of Autophagy in Skin Pigmentation Abnormalities

Autophagy is an essential cellular process for homeostasis accomplished by degrading and recycling damaged cellular components. Autophagic activity declines with aging [136]. However, defects in autophagy can also cause skin aging with an accumulation of damaged cellular components [137]. The association between reduced autophagy and skin aging has mostly been investigated in photoaging. The phenotype of skin aging induced by UVA irradiation in human fibroblasts and mice with Cockayne syndrome type B deficiency is accompanied by reduced autophagic activity and is rescued by the improvement in autophagic function [138]. Repeated UVA irradiation results in the accumulation of autophagosomes and lysosomes, suggesting that photoaging can reduce autophagic activity at the degradation stage [139,140]. Oxidative stress-induced cellular senescence in murine melanocytes and keratinocytes associated with autophagy-related 7 *(Atg7)* deficiency [12,141] and the protective role of caffeine in oxidative stress-induced senescence of human epidermal keratinocytes through autophagy induction [142] suggest that autophagy defects can accelerate skin aging by increasing oxidative stress.

#### 3.5.1. Autophagy and Skin Hypermelanosis

The melanosome is a lysosome-related organelle. Autophagy-related regulators are involved in the formation, maturation, and degradation of melanosomes in melanocytes [143]. In vitro and in vivo results have demonstrated the association between autophagy and skin pigmentation abnormalities (Table 6).

The diversity of skin color was affected by epidermal heat shock 70 kDa protein 1A (Hsp70-1A), which modulates autophagic melanosome degradation in keratinocytes [144]. Accelerated keratinocyte senescence by arginase-2 upregulation impairs autophagy and reduces melanosome degradation, resulting in hyperpigmentation in melasma [145]. Association between premature skin aging with autophagy reduction was also detected in the hyperpigmented lesions of patients with senile lentigo, whereas autophagy activation ameliorated pigmentation in ex vivo lesional skin [17]. Chemical agents such as tranexamic acid, β–mangostin, 3-*O*-glyceryl-2-*O*-hexyl ascorbate, and Melasolv induced skin lightening through autophagy activation [146,147,148,149]. Light-emitting diodes can also activate autophagy, resulting in skin lightening [150].

#### 3.5.2. Autophagy and Skin Hypomelanosis

Recessive mutations in *EPG5* encoding the ectopic P-granules autophagy protein 5 homolog, a key regulator of autophagy, cause Vici syndrome. As one of the most typical examples of congenital autophagy disorders, patients with the syndrome exhibit characteristic features that include oculocutaneous hypopigmentation [151]. Although the mechanism of hypopigmentation remains to be clarified, defective autophagosome-lysosome fusion in EPG5-depleted melanocytes [152] may disrupt the antioxidant defense system. Autophagic dysregulation by activation of the mTOR signaling pathway in melanocytes has been identified as a mechanism of reduced pigmentation in hypopigmented macules in patients with tuberous sclerosis complex (*TSC*) gene mutations [153,154]. The loss of glycoprotein nonmetastatic melanoma protein B (GPNMB) causes amyloidosis cutis dyschromica, which is characterized by generalized hyperpigmentation mottled with hypopigmented macules [155]. GPNMB is a melanocytic cell marker involved in multiple functions, including melanosome formation and autophagy [156]. GPNMB knockdown reduced the number of melanosomes in melanocytes [157].

GPNMB has also been downregulated in vitiligo keratinocytes, causing the loss of melanocytes [158]. The finding indicates the regulatory role of GPNMB in skin hypomelanosis by reducing melanocyte survival in addition to reducing melanogenesis.

## 4. Conclusions

Skin diseases with pigmentation abnormalities are difficult to treat primarily due to unknown causes or pathogenesis. The pathomechanisms in individual patients can be different even in the same disease. It is important to identify the cause to manage skin pigmentation abnormalities. To propose the possible role of skin aging in abnormal pigmentation, the association between the identified mechanisms involved in skin aging and skin pigmentation abnormalities in various inherited and acquired disorders were reviewed. The mechanisms implicated in skin aging include oxidative stress, which is the most pivotal cause of skin aging, DNA damage, telomere shortening, decreased melatonin, and autophagy impairment. Both skin aging and pigmentation abnormalities in various inherited disorders caused by DNA damage or telomere shortening indicate the relevant relationship between skin aging and pigmentation abnormalities. However, other mechanisms may not yet sufficiently support the relationship between skin aging and pigmentation abnormalities (Figure 2). Epigenetic changes, particularly DNA methylation [159,160] and microRNAs [161], are also proposed to be involved in skin aging without an identified role in skin pigmentation. More studies are needed to prove the reliable role of skin aging in various conditions showing abnormal skin pigmentation.

## Figures and Tables

**Figure 1 ijms-22-03727-f001:**
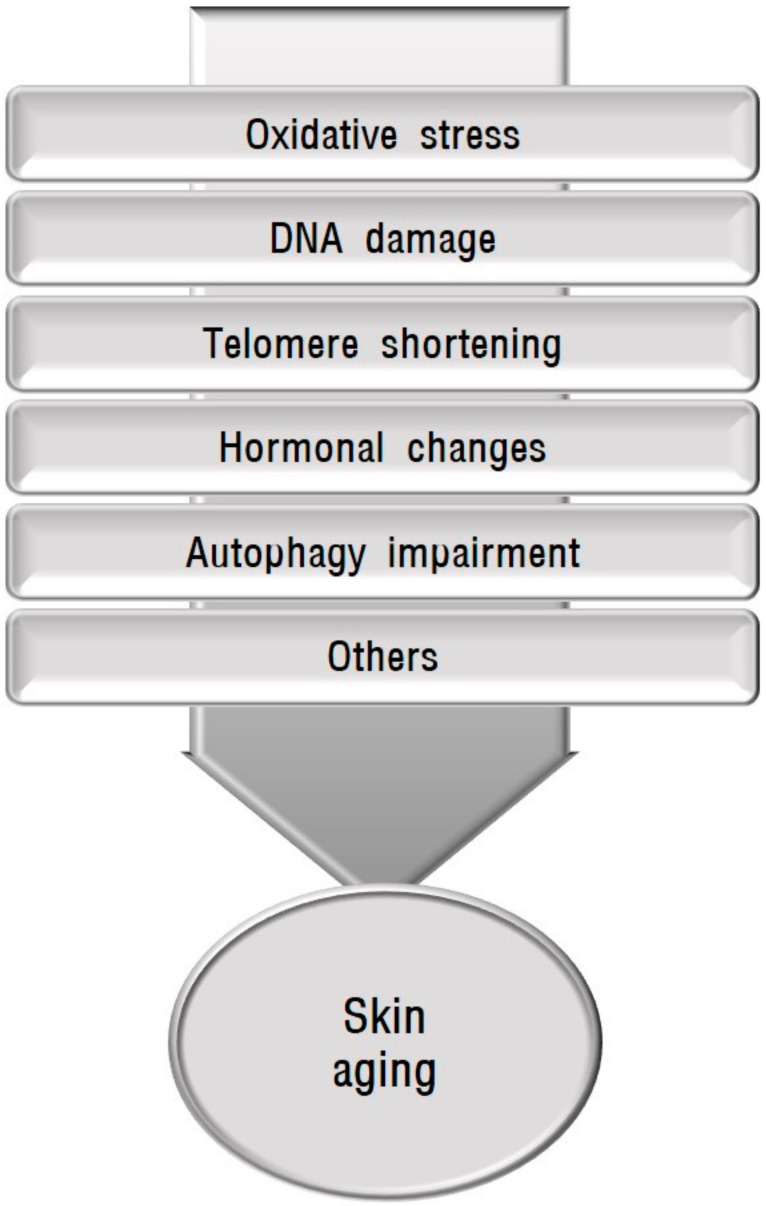
Schematic view of causative factors involved in skin aging. Skin aging is influenced by several factors including oxidative stress, mitochondrial DNA mutations, DNA damage, telomere shortening, hormonal changes, and autophagy impairment.

**Figure 2 ijms-22-03727-f002:**
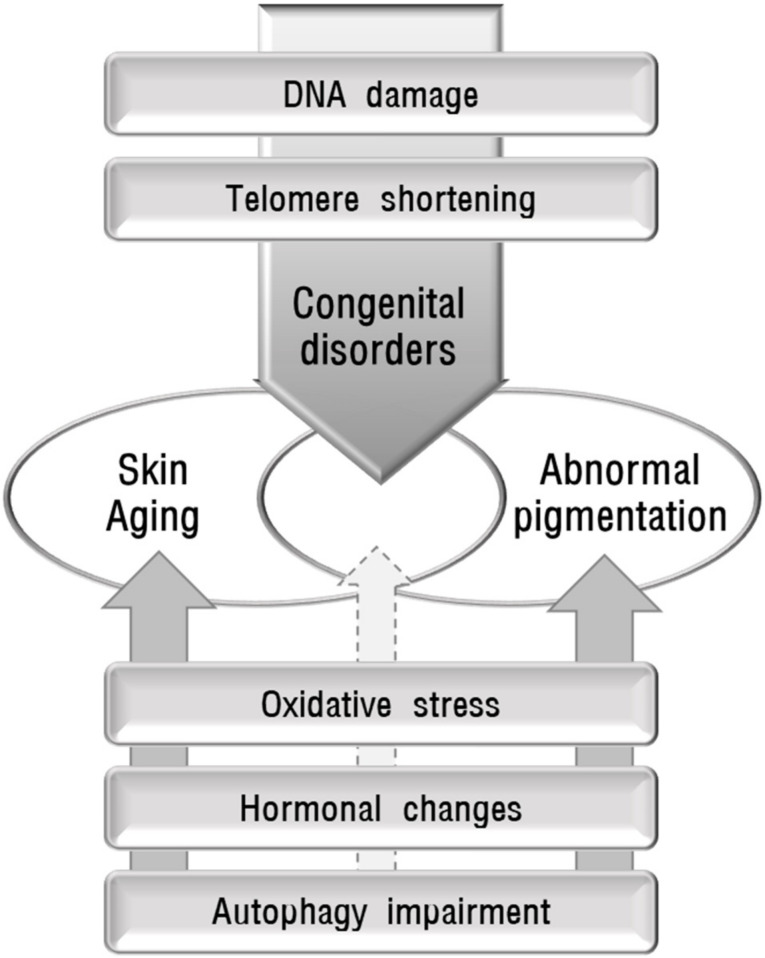
Schematic view of the possible association between abnormal pigmentation and skin aging. Some causative mechanisms, such as DNA damage and telomere shortening, are involved in hereditary disorders that show accelerated aging and skin hyperpigmentation or hypopigmentation, providing reliable evidence for the association between skin aging and pigmentation abnormalities. However, oxidative stress, hormonal changes, and autophagy impairment, may not yet sufficiently support the relationship between skin aging and pigmentation abnormalities.

**Table 5 ijms-22-03727-t005:** Abnormal skin pigmentation in congenital disorders related to telomere shortening.

Pigment Change	Mutated Gene	Dysfunction	Skin Symptom	Reference
Hyper	*DKC1*	Telomere maintenance defect	Reticular hyperpigmentation, Oral leukoplakia, Nail dystrophy	[116,117,118]
Hyper and hypo	*XPC*	Defect in telomere stability and recognition of DNA photoproducts	Photosensitivity, Lentigines, Hyperpigmentation and hypopigmentation, Accelerated photoaging	[99,117]
	Fanconi anemia	Defect in DNA repair for telomere maintenance	Poikilodermatic change with hypopigmentation, hyperpigmentation, and telangiectasia	[100,101]
	*LMNA*	Rapid telomere erosion	Hyperpigmentation and hypopigmentation over areas of sclerodermoid change	[95,119,120,121]

**Table 6 ijms-22-03727-t006:** Abnormal skin pigmentation in disorders related to autophagy impairment.

Pigment Change	Causative Factor	Dysfunction	Skin Symptom	Reference
Hyper	Reduced Hsp70-1A	Decrease in autophagic melanosome degradation	Skin color	[144]
	Accelerated keratinocyte senescence caused by arginase-2 upregulation	Decrease in autophagic melanosome degradation	Hyperpigmented lesion of melasma	[145]
	Premature skin aging	Significant decreases in autophagy	Hyperpigmented lesion of senile lentigo	[17]
	Chemical agents	Autophagy activation	Lightening of skin hyperpigmentation	[146,147,148,149]
	Light-emitting diodes	Autophagy activation	Lightening of skin hyperpigmentation	[150]
Hypo	*EPG5* mutations	Autophagosome-lysosome fusion defect, leading to antioxidant defense system disruption	Oculocutaneous hypopigmentation	[151,152]
	*TSC* mutations	Autophagic dysregulation by mTOR signaling pathway activation	Hypopigmented macules	[153,154]
	Downregulated GPNMB	Defect in melanosome formation and autophagy	Hypopigmented macules	[155,156,157]
		Reduced melanocyte survival	Vitiligo	[158]

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
