# Peer review of "Skin Pigmentation Abnormalities and Their Possible Relationship with Skin Aging"

_ijms, 2021, doi:10.3390/ijms22073727_

Round 1

Reviewer 1 Report

The manuscript entitled "Aging-related Skin Pigmentation Abnormalities" describes the pigmentation disturbances and some molecular mechanisms. However, the relationship between these pigmentation abnormalities and age-related mechanisms is not clearly indicated and noticeable. Thus the title does not fit the manuscript content.

In addition, readers would appreciate more information on melanin synthesis and its regulation (see: Physiol Rev. 2004 Oct;84(4):1155-228. doi: 10.1152/physrev.00044.2003)

Author Response

As for the recommendation of Reviewer 1;

The manuscript entitled "Aging-related Skin Pigmentation Abnormalities" describes the pigmentation disturbances and some molecular mechanisms. However, the relationship between these pigmentation abnormalities and age-related mechanisms is not clearly indicated and noticeable. Thus the title does not fit the manuscript content. 

:The title is changed as <Skin Pigmentation Abnormalities and their Possible Relationship with Skin Aging> with reference to the suggestion of another reviewer.

In addition, readers would appreciate more information on melanin synthesis and its regulation (see: Physiol Rev. 2004 Oct;84(4):1155-228. doi: 10.1152/physrev.00044.2003) 

: The information on melanin pigmentation with melanin synthesis and its regulation is briefly described with the references including your recommended reference under the newly formed small title <Melanin pigmentation and its abnormalities accompanied with skin aging>.

Reviewer 2 Report

This is a very interesting, well-written, comprehensive review on pigmentation-related skin diseases of the potential to affect skin aging. I've got only a few remarks, which may improve the overall impression given by this paper. 

  1. Despite it being mentioned several times in the text and implied by the figures, I cannot find a clear relationship and convincing presumption that these skin disorders are really aging-related. It is really difficult to dissect whether the disorders are aging-related or perhaps aging is disease-related. Numbers of them are of genetic background. There is also a lack of clear pictures of pigmentation-not related symptoms of skin aging. I, therefore, suggest changing the title to "Pigmentation-related skin disorders and their possible relationships with skin aging", or similar.
  2. A strong factor of skin aging is melanin itself, melanogenesis, and melanin degradation or photodegradation. There could be a small chapter entitled "melanogenesis and its implication in skin aging". The aspect of eu- vs. pheomelanin could be especially exposed in this place.

Author Response

Despite it being mentioned several times in the text and implied by the figures, I cannot find a clear relationship and convincing presumption that these skin disorders are really aging-related. It is really difficult to dissect whether the disorders are aging-related or perhaps aging is disease-related. Numbers of them are of genetic background. There is also a lack of clear pictures of pigmentation-not related symptoms of skin aging. I, therefore, suggest changing the title to "Pigmentation-related skin disorders and their possible relationships with skin aging", or similar.

: According to your suggestion, the tile is changed as <Skin Pigmentation Abnormalities and their Possible Relationship with Skin Aging>.

 A strong factor of skin aging is melanin itself, melanogenesis, and melanin degradation or photodegradation. There could be a small chapter entitled "melanogenesis and its implication in skin aging". The aspect of eu- vs. pheomelanin could be especially exposed in this place.

: I appreciate your comments. Melanin pigmentation with protective effect of eumelanin, but not pheomelanin on UV-induced damage including skin aging is described. The content about change in melanin pigmentation from melanin synthesis and degradation with relation to aged human skin, despite not many, is added under the subtitle <Melanin pigmentation and its abnormalities accompanied with skin aging>.

Round 2

Reviewer 1 Report

Author addressed my previous comments and corrected the manuscript.